# Large-Scale Pretraining unlocks
# Few-Shot Prediction for Relational Data

## Abstract

Few-shot adaptation has been crucial to the success of foundation models for language and vision, but foundation models for structured relational data still require thousands of labeled examples to perform well. We show that, when pretrained at scale with the right recipe, a Relational Transformer (RT) becomes a strong few-shot predictor across diverse databases. Our recipe has three ingredients: THE JOIN, the largest open relational pretraining corpus to date, with 6,255 forecasting tasks across 650 real-world databases; a pretraining procedure that mixes context sizes, masks many cells per window, and fills the context via random-walk retrieval; and test-time compute scaling via context ensembling and per-task context tuning. On RelBench, the resulting model matches the strongest in-context learning pipelines (RDBLearn + TabICLv2 and an LLM Agent + TabICLv2) using 23–32× fewer in-context labels, and exceeds the prior fully-supervised state of the art (RelGNN) without any task-specific training. Ablations show that schema semantics, multi-cell masking, random-walk retrieval, and mixed task pretraining each contribute materially to the regime.

## 1. Introduction

Foundation models have changed how machine learning systems are adapted to new tasks (Bommasani et al., 2022). In language and vision, pretrained models can solve new tasks from only a small number of in-context examples (Brown et al., 2020; Alayrac et al., 2022). This few-shot paradigm remains largely out of reach for relational data, even though relational databases hold much of the world's structured enterprise, scientific, and operational data (Fey et al., 2024; Vo-

[1]Anonymous Institution, Anonymous City, Anonymous Region, Anonymous Country. Correspondence to: Anonymous Author <anon.email@domain.com>.

Preliminary work. Under review by the International Conference on Machine Learning (ICML). Do not distribute.

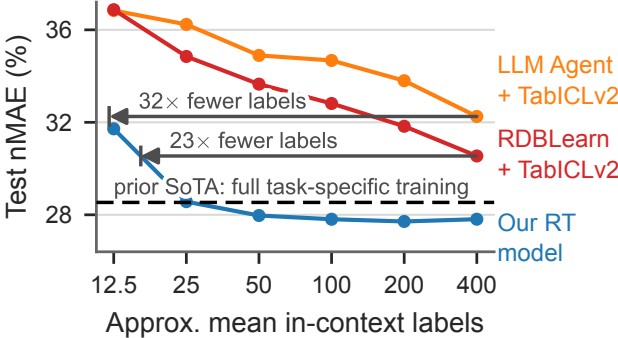

*Figure 1.* Z-score nMAE for in-context regression averaged over the 9 RelBench tasks held out from pretraining. For any point on the $x$-axis, all curves see the same number of labeled rows on average (denoted "Approx." since the actual count varies by task). Our pretrained Relational Transformer (RT) matches the strongest in-context learning baselines using ≈23–32× fewer labels and surpasses the prior fully-supervised state of the art (RelGNN (Chen et al., 2025)).

gel et al., 2022). Strong relational predictors today typically require task-specific training (Robinson et al., 2024; Chen et al., 2025; Dwivedi et al., 2025a; Peleška & Šír, 2024), substantial labeled data (Fey et al., 2025; Hudovernik et al., 2026), or careful feature engineering (Chen & Guestrin, 2016; Shwartz-Ziv & Armon, 2022). This limits predictive modeling exactly where it would be most useful: when labels are scarce, tasks change frequently, or users need rapid ad-hoc predictions.

Recent work has made progress toward foundation models for relational prediction, but existing approaches remain mismatched to few-shot deployment. Tabular foundation models such as TabPFN (Hollmann et al., 2023; 2025) and TabICLv2 (QU et al., 2025; Qu et al., 2026) provide in-context learning for single-table prediction; systems such as RDBLearn (Zhang et al., 2026) extend them to relational databases by aggregating relational neighborhoods into engineered features. Flattening the database in this way removes the relational structure needed to reason over entities, links, and multi-table dependencies. Schema-agnostic relational foundation models, including Griffin (Wang et al., 2025), KumoRFM (Fey et al., 2025; Hudovernik et al., 2026), and the Relational Transformer (RT) (Ranjan et al., 2026), pretrain across databases and adapt to new schemas at inference

time, but have not yet been optimized for few-shot behavior.

We argue that this label inefficiency is not primarily an architectural limitation but a mismatch between existing relational foundation-model recipes and the demands of few-shot prediction. RT is a natural starting point: it operates directly over database cells, schema metadata, and relational links, so labeled task rows, unlabeled rows, and linked evidence all live in a single context window. However, the original RT recipe was not designed to turn the architecture into a strong in-context learner. Three gaps are central. *First*, public relational pretraining data remains narrow, leaving rare-event, cold-start, and heavy-tailed regimes sparsely represented (Gu et al., 2026; Motl & Schulte, 2015). *Second*, existing masked-cell pretraining provides sparse supervision at a single short context length, while few-shot inference relies on long contexts containing many labeled examples. *Third*, standard context construction either ignores relational structure or expands local neighborhoods naively, without preferentially retrieving rows that are both relevant to the target and useful as supervision.

**Contributions.** We close these three gaps and show that a pretrained RT becomes a strong few-shot predictor when its data, training, retrieval, and test-time compute recipe are jointly designed for this regime (Fig. 1). **(1)** We assemble THE JOIN, the largest open pretraining corpus of relational data to date: 6,255 forecasting tasks across 650 real-world databases from 63 domains, with deliberate coverage of rare-event, cold-start, and heavy-tailed regimes. **(2)** We introduce a pretraining recipe that aligns training with few-shot inference, combining mixed context sizes, multi-cell masking, and a random-walk-based context retriever. **(3)** We study context ensembling and per-task context tuning as two complementary forms of test-time compute. On RelBench, our model matches RDBLearn + TabICLv2 and LLM Agent + TabICLv2 using $23\times$ and $32\times$ fewer in-context labels, reaches the prior fully-supervised SoTA on RelBench regression nMAE with $\sim24$ labels, and surpasses it with $\sim46$, all without task-specific training.

## 2. Background

The Relational Transformer (RT) (Ranjan et al., 2026) is the backbone we build on. RT tokenizes a relational database at the *cell* level: each cell is a token whose embedding combines its value, a datatype-specific encoding, and frozen text embeddings of its table and column names. Feature, foreign-key, temporal, and target cells share one representation space, so the model is independent of any fixed schema. Each transformer block applies three structured attention layers: *column attention* (cells in the same column), *feature attention* (cells in the same row), and *neighbor attention* (cells in rows connected by primary–foreign-key links). Predictive tasks are expressed as masked-cell prediction: a

forecasting target becomes a single masked cell on a task-table row, predicted from the surrounding unmasked context. The original RT was pretrained with single-cell masking at a fixed short context length and was not optimized for few-shot inference.

## 3. THE JOIN: A Large-Scale Pretraining Corpus

Existing relational benchmarks such as RelBench (Robinson et al., 2024; Gu et al., 2026) and the CTU repository (Motl & Schulte, 2015) comprise small, narrowly-scoped collections that miss the rare-event, cold-start, and heavy-tailed regimes relational foundation models need to see at pretraining time. We introduce THE JOIN, a corpus of 650 real-world relational databases from five disjoint source classes (curated repositories, public APIs, domain portals, community dumps, and benchmarks) spanning 63 domains, together with 6,255 forecasting tasks generated by a four-stage pipeline: (i) **collection** of raw databases; (ii) **standardization** (boolean/numeric coercion, foreign-key inference, temporal normalization, entity-axis subsampling, and source-overlap exclusion to prevent leakage into RelBench); (iii) **task generation** by an LLM proposer over 25 template families spanning eight reasoning regimes (aggregation, threshold/existence, trend/change, behavior/churn, ranking, adoption/novelty, streak, and tail-event), with deterministic validation; and (iv) **task filtering** that retains *learnable* and *random-tier* tasks via an XGBoost score, discarding trivial, drift, and degenerate ones. The pipeline is driven end-to-end by Claude Code agents calling deterministic checkers, which makes corpus breadth of this scale tractable. Full details are in § C and Tab. 3.

## 4. Pretraining Recipe

Given a context size $L$, RT subgraph-samples $L$ cells around a target row and trains by predicting masked cells. Three changes turn this into an effective few-shot pretraining setup.

**Mixed context sizes.** Larger $L$ improves prediction but is more expensive, and the same model should handle small and large $L$ at test time so that inference compute can be traded against quality. We sample $L$ uniformly from a discrete set $\mathcal{L}$ per training batch, keep the per-step batch size fixed, and use gradient accumulation adaptively to keep $L \times B$ memory bounded.

**Random-walk retrieval.** The original RT samples context via a bounded-width breadth-first search (BFS) over primary–foreign-key links from the target row; as $L$ grows, BFS exhausts the local neighborhood and fills the budget with low-relevance rows. We use a two-stage scheme. *Stage*

*1:* run $W$ random walks of length $K$ from the target (each step restricted to PK–FK neighbors with timestamp $\leq t^\star$ to avoid temporal leakage), scoring each candidate task-table row by its visit count (with optional recency tie-break controlled by $\rho$). *Stage 2:* consume the top-scoring rows in order as seeds for a local BFS with per-seed cell budget $\ell$ and width $w$ until the total budget $L$ is filled. $(\ell, w, \rho)$ are exposed as test-time hyperparameters and randomized during training so the model is robust across settings. Unlike vector-similarity retrievers, random-walk retrieval needs no precomputation and scales naturally to large database corpora. Full procedure in Algorithm 1.

**Multi-cell masking.** The original RT masks only the target cell, so the supervision-to-compute ratio worsens as $L$ grows. Following Dong et al. (2025), for each context $i$ we draw a per-context mask rate $p_i \sim \text{Unif}([0, p_{\max}])$, then independently mask each non-target, non-missing cell with probability $p_i$. The hierarchical Bernoulli scheme yields dense per-window supervision and exposes the model to relational structure at multiple masking scales.

## 5. Pretraining Results

**Pretraining details.** We pretrain an 85M-parameter RT (12 layers, hidden 512, 8 heads) at BFloat16 on $32\times$H100 GPUs for 2 days, following the architectural improvements of Kothapalli et al. (2026) over the original RT (Ranjan et al., 2026). We sample $L \sim \text{Unif}\{1k, 2k, 4k, 8k\}$, $\ell \in \{512, 1k, 2k\}$, $w \in \{16, 32, 64, 128\}$, $\rho \in \{0, 1\}$, with $W{=}10{,}000$ walks of length $K{=}20$ and $p_{\max}{=}0.5$. Optimization runs for 100k steps at global batch size 1024, using Muon (Jordan et al., 2024) for 2D weights and AdamW otherwise, at constant LR $5\times10^{-4}$ after warmup. We track an SWA-EMA (momentum 0.9995) and select the best EMA checkpoint on validation (Tian et al., 2025; Ranjan et al., 2024), picking checkpoints independently for regression and classification.

**Baselines.** **RDBLearn** (Zhang et al., 2026) aggregates relational neighborhoods into a flat feature table and applies a tabular predictor. An **LLM Agent** (Claude Code with Opus 4.6) explores the database and writes SQL feature-engineering queries before doing the same. We pair each with two predictors: **TabICLv2** (QU et al., 2025; Qu et al., 2026), an in-context tabular foundation model; and a **Ridge** regressor/classifier. We also include the **original RT** (Ranjan et al., 2026) as a relational baseline. Baselines see the same task rows as our RT (extracted from the same retrieved context) but are unconstrained on feature processing.

**Experimental setup.** We evaluate on RelBench (Robinson et al., 2024) (21 forecasting tasks: 9 regression, 12 binary classification over 7 databases *held out from pretraining*). For every task we sweep the test-time context size $L \in \{256, 512, 1k, 2k, 4k, 8k\}$ cells with all other context hyperparameters fixed, ensuring that the 256-cell context is a subset of the 512-cell context, and so on. We report Z-score normalized MAE (nMAE: MAE divided by the train-target standard deviation; lower is better) for regression and AUROC (higher is better) for classification, averaged uniformly within each category. Since we sweep over RT context sizes and task-table rows are encountered organically during context sampling, the exact in-context label count differs across tasks; we therefore report the mean over tasks and plot against a conservative upper bound on the $x$-axis ("Approx." in Figs. 1 and 2). All hyperparameter and checkpoint selection uses the validation split; numbers are reported on the held-out test split.

**Observations.** Fig. 2 shows our model dominates every baseline at every label count on both metrics, by up to 18% relative nMAE on regression and 3% relative AUROC on classification over the next-best pipeline. The gap with the TabICLv2 pipelines is largest at low label counts and narrows as more labels arrive—the few-shot regime is precisely where RT helps most. At $L{=}8k$ cells ($\approx360$ labels on average), our model also surpasses the prior fully-supervised SoTA on RelBench regression nMAE (RelGNN (Chen et al., 2025), trained per task on the full label budget). The original RT (Ranjan et al., 2026) fails to learn from more in-context labels as $L$ grows (regression nMAE *worsens* at long contexts), so the gap between it and our model directly quantifies the impact of the data and recipe. With $\sim400$ labels and no task-specific training, our context-tuned model also reaches the per-task quality of RDBLearn (Zhang et al., 2026) trained on up to 10,000 labels per task (Tab. 2; numbers from Hudovernik et al. (2026)).

## 6. Test-Time Compute Scaling

### 6.1. Context ensembling

Both retriever stages are stochastic, so different sampler seeds yield equally-viable contexts. Averaging predictions over $n_{\text{ens}}$ such contexts adds no pretraining cost, parallelizes trivially, and gains up to 3.2% relative nMAE and 4.3% relative AUROC at $n_{\text{ens}}{=}16$ (Fig. 7). It is complementary to context size: 16 seeds at $L{=}512$ beat 1 seed at $L{=}8k$ on classification despite using less inference compute, since RT is quadratic in $L$ but linear in $n_{\text{ens}}$.

### 6.2. Context tuning

The default uses a single shared $(\ell, w, \rho)$ across tasks. Sweeping the 32 configurations on validation at $L{=}8k$ and selecting the best per task improves classification at every $L$ ($+1.2\%$ to $+3.8\%$ AUROC) and yields a 3.8% nMAE gain at the tuning point (Fig. 8). The regression gain only materi-

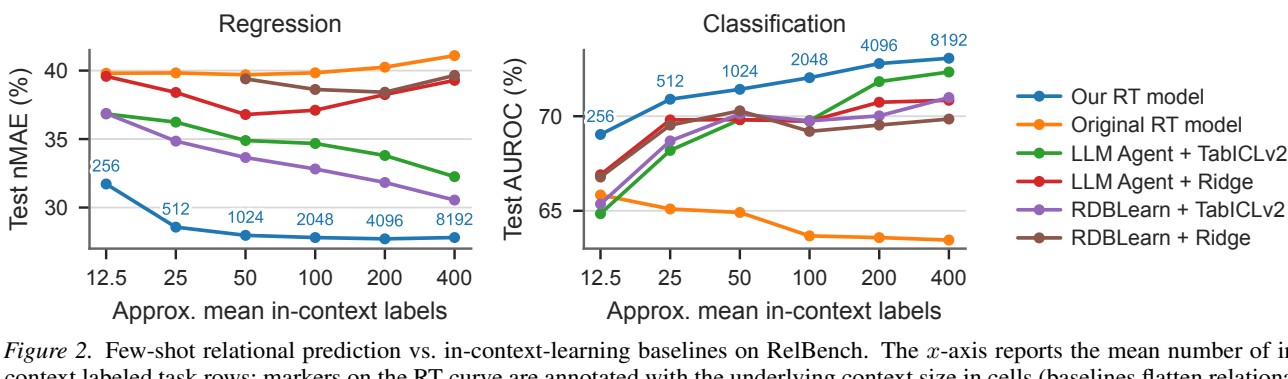

*Figure 2.* Few-shot relational prediction vs. in-context-learning baselines on RelBench. The $x$-axis reports the mean number of in-context labeled task rows; markers on the RT curve are annotated with the underlying context size in cells (baselines flatten relational neighborhoods and consume the same labels differently). Our model (blue) dominates every baseline at every $L$, by up to $18\%$ relative nMAE on regression and $3\%$ relative AUROC on classification over the next-best baseline. "Original RT model" is from (Ranjan et al., 2026).

alizes once $L$ is large enough to host the chosen $\ell$, so $(\ell, w)$ tuned at long contexts does not transfer unconditionally to short ones.

# 7. Ablation Studies

## 7.1. Alternatives to random-walk retrieval

We compare random-walk retrieval against five alternatives: uniform random task-row selection; BFS of width 32 and 256 rooted at the target (the original recipe of Ranjan et al. (2026)); and two vector-similarity retrievers that score task rows via RDBLearn features or RT embeddings. Random-walk retrieval is the strongest variant on regression at every $L$ and on classification at $L=8k$. Its advantage over wide BFS peaks at intermediate $L$ ($\approx 3.8$ nMAE at $L=512$) and narrows at large $L$ as BFS eventually exhausts the relevant subgraph; against structurally-naive retrievers (uniform random, embedding similarity) the gap *widens* with $L$ (4–8 nMAE at $L=8k$): these alternatives stall while random walk and BFS keep improving. Higher label yield alone does not suffice: RDBLearn-feature similarity delivers more labels at every $L \geq 512$ (464 vs. 385 at $L=8k$ on regression) yet trails by $\sim 4$ nMAE—random walk wins by surfacing rows that are *both* topologically close to the target *and* label-bearing.

## 7.2. Multi-cell masking rate

Multi-cell masking gains regression monotonically in $p_{\max}$: $p_{\max}=0.5$ beats single-cell masking ($p_{\max}=0$) by $4.2\%$ relative nMAE at $L=8k$, with classification separating at $L \geq 1k$. RT's cell-level tokenization is what enables this dense per-window supervision: row-level models (QU et al., 2025; Qu et al., 2026; Wang et al., 2025) cannot exploit it.

## 7.3. Schema semantics

Language-model embeddings of table/column names deliver a consistent $6.7\%$–$9.1\%$ nMAE and $2.6\%$–$4.1\%$ AUROC

| Configuration | nMAE(%)↓ | AUROC(%)↑ |
|---|---|---|
| Prior RT (Ranjan et al., 2026) | 41.1 (+13.3) | 63.4 (−9.7) |
| **Ours, full recipe** | **27.8** | **73.1** |
| − schema semantics | 29.8 (+2.0) | 70.7 (−2.4) |
| − multi-cell masking | 29.5 (+1.7) | 71.6 (−1.5) |
| − random-walk retrieval | 29.1 (+1.3) | 72.3 (−0.8) |
| − mixed pretraining | 28.2 (+0.4) | 72.3 (−0.8) |
| + per-task context tuning | 26.8 (−1.0) | 74.9 (+1.8) |
| + 16-seed context ensembling | 27.4 (−0.4) | 74.8 (+1.7) |

*Table 1.* Recipe summary on RelBench at $L=8k$ (averages over the 9 regression and 12 classification tasks; ablations at 32k steps, full recipe at 100k; deltas from the full recipe in parentheses; − removes, + adds).

gain over a control where the names are randomly shuffled, with the gap stable across $L$. Schema text supplies signal that does not collapse as labels arrive—precisely the signal that synthetic-data-only tabular foundation models (QU et al., 2025; Qu et al., 2026; Kothapalli et al., 2026) cannot recover.

## 7.4. Pretraining task mix

Forecasting-only pretraining loses $2.3\%$–$4.2\%$ nMAE on regression; autocompletion-only is tied on regression but loses up to $3.5\%$ AUROC at large $L$. Only the mixed objective is competitive on both metrics.

# 8. Conclusion

We introduce a few-shot pretrained Relational Transformer that predicts from *hundreds* of in-context labels rather than the tens of thousands prior relational predictors require, by pairing THE JOIN with mixed context sizes, multi-cell masking, random-walk retrieval, and test-time context ensembling and tuning. Without task-specific training, it beats every in-context baseline and the prior fully-supervised SoTA on RelBench.

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

## A. Related Work

**Relational deep learning and benchmarks.** Relational deep learning operates directly on multi-table databases by representing them as heterogeneous relational entity graphs: rows are nodes, tables define node types, and primary–foreign-key links define typed edges (Fey et al., 2024; Robinson et al., 2024). Graph neural networks propagate information across linked rows and tables, enabling end-to-end prediction over multi-table data (Robinson et al., 2024; Chen et al., 2025). Recent work has improved the efficiency and expressiveness of relational message passing (Chen et al., 2025), and transformer-based relational architectures have also been proposed (Peleška & Šír, 2024; Dwivedi et al., 2025a). Kanatsoulis et al. (2025) introduces an efficient positional-encoding framework based on lightweight message passing that provides structural features useful for relational deep learning models. Broader surveys frame these models as part of a larger agenda for relational deep learning (Dwivedi et al., 2025b). Benchmarks have been central to progress: RelBench (Robinson et al., 2024) introduces a standardized suite of relational prediction tasks, while 4DBInfer (Wang et al., 2024) and RelBenchV2 (Gu et al., 2026) expand scale and diversity. All of these models, however, are typically trained per-task and per-schema, which is the gap that schema-agnostic foundation models aim to close.

**Relational foundation models.** A complementary line seeks foundation models that transfer across schemas and tasks rather than training a separate predictor per database. KumoRFM (Fey et al., 2025) and KumoRFM-2 (Hudovernik et al., 2026) study in-context learning for relational data at scale. Griffin (Wang et al., 2025) develops a graph-centric relational foundation model. Beyond the relational setting, recent graph foundation models scale these ideas across heterogeneous graphs at billion-parameter scale (Bechler-Speicher et al., 2026). The Relational Transformer (RT) (Ranjan et al., 2026) takes a complementary cell-level approach: database cells are tokens, schema semantics are part of the input, and relational attention masks encode primary–foreign-key structure. PLUREL (Kothapalli et al., 2026) pretrains RT on synthetic multi-table data, enabling controlled studies of synthetic-data scaling. Our work builds directly on RT because its cell-level formulation is natural for few-shot relational prediction: labeled task rows, unlabeled rows, schema metadata, and linked evidence all live in the same context window, making masked-cell prediction a single interface for both pretraining and inference. The original RT work primarily establishes the architecture; it does not develop a recipe specifically designed to unlock few-shot learning, which is the gap we close.

**Tabular foundation models and relational tabularization.** TabPFN and its successors show that transformers can perform strong in-context prediction when labeled rows are provided in context (Hollmann et al., 2023; 2025). More recent models—TabICLv2 (QU et al., 2025; Qu et al., 2026), PORTAL (Spinaci et al., 2024), and CARTE (Kim et al., 2024)—extend the paradigm to larger datasets and richer feature spaces. Relational data, however, contains structure that single tables do not: predictive evidence may lie several joins from the target. Tabular foundation models can be applied here only after flattening: RDBLearn (Zhang et al., 2026) aggregates relational neighborhoods into engineered features and then applies a tabular in-context learner. This is an effective baseline, but the flattening step changes the input representation, so the model no longer reasons over cells, schemas, timestamps, foreign-key links, or multi-table structure.

**Large language models and graphs.** A separate thread applies LLMs to structured/relational data by serializing tables or database neighborhoods as text (Wydmuch et al., 2024). This is convenient but inefficient and brittle: relational structure must be expressed as text, long database contexts can exceed the context window, and the resulting input format differs from language pretraining. Hybrid graph/relational + LLM approaches partially address these issues but often still depend on task-specific components (Wu et al., 2025). On the graph side, recent work has built foundation models for graphs at scale: MoleBERT (Xia et al., 2023) pretrains GNNs with masked-atom prediction; Beaini et al. (2024) scale molecular pretraining with billions of labels; PRODIGY (Huang et al., 2023) introduces prompt-based graph representations for in-context learning; GraphAny (Zhao et al., 2025a) studies zero-shot node classification with disjoint feature/label spaces; ULTRA (Galkin et al., 2024) learns transferable relational representations for zero-shot reasoning over knowledge graphs; GraphGPT (Zhao et al., 2025b) converts graphs into reversible token sequences for generative pretraining; and Bechler-Speicher et al. (2026) scales the graph-foundation-model paradigm to billion-scale heterogeneous graphs with link prediction pretraining.

## B. Limitations and Future Work

Our evaluation centers on the 21 forecasting tasks of RelBench, which leaves open how the recipe transfers to the harder distributional clusters that THE JOIN covers but no public benchmark exposes. We expect the advantage to grow in those regimes but do not measure it here. The model handles a single task per inference call, whereas production workloads often

predict many targets jointly. We do not study fine-tuning on top of the few-shot model, which is the natural next step when more labels are available.

We highlight three directions as most promising for future work. First, further scaling both the corpus and the model. Our classification AUROC continues to improve at $L = 8,192$ and our regression nMAE plateaus around $L \approx 512$, suggesting that classification will benefit most from longer contexts while regression likely needs stronger models or richer pretraining to improve further. Second, integrating the few-shot pretrained Relational Transformer with downstream tooling, including supervised fine-tuning, multi-task prediction heads, and SQL-style query interfaces. Third, extending the recipe beyond entity-level forecasting to link prediction and cross-database transfer, both of which the cell-level masked objective should accommodate without architectural changes.

## C. Additional Details on THE JOIN

**Agentic execution.**    Constructing THE JOIN manually would require months of engineering, particularly for web collection (each source has its own download mechanism, format, schema, parsing failures, and quality issues). We use Claude Code agents as the execution layer for corpus construction. Agents identify candidate sources, implement and debug domain-specific collectors, adapt standardization routines to newly observed edge cases, run the pipeline, inspect logs and intermediate artifacts, and iterate until each stage reaches its completion criteria. Validation, scoring, filtering, baseline computation, and provenance tracking are performed by deterministic scripts based on human-defined criteria. This separation preserves auditability: numerical claims are derived from mechanical outputs, while human supervision is focused on reviewing code, redirecting decisions, and approving released artifacts.

**Phase 1 – Collection.**    Five disjoint source classes: (1) curated relational repositories (CTU, OGB, OpenAlex), (2) public APIs (US Treasury, SEC EDGAR, CFPB, FDA, World Bank, AACT, EIA), (3) domain portals (Lahman, ATP, StatsBomb, Cricsheet, OpenFlights, Stooq, MusicBrainz), (4) community dumps (Stack Exchange, GitHub-hosted datasets), and (5) benchmarks (Kaggle, Spider/BIRD). Every contributing source is either a bulk archive published under a permissive license or a public API whose documented terms permit programmatic research use. Collectors download and parse the data in its native format, recover table and key metadata when available, and convert to a common parquet representation. Per-collector normalization stores primary-key, foreign-key, and time-column metadata alongside each parquet table. As a preliminary quality filter, we admit databases with at least two tables joined by a foreign key and at least one parseable timestamp column. Tab. 3 lists the 63 collected domains.

**Phase 2 – Standardization.**    Six idempotent detector-and-transformer fixes recover canonical relational structure: (1) *boolean coercion*, rewriting string and `uint8` columns whose values fall in a canonical truth-set as Arrow booleans; (2) *numeric coercion*, casting string-encoded numerics to ints or floats with an identifier blacklist guarding zip codes and ISBNs; (3) *foreign-key inference* by name match plus an $\geq 80\%$ value-containment test against parent primary keys; (4) *temporal normalization*, electing a single time column per table from regex, name-hint, and integer-year candidates; (5) *entity-axis subsampling* of oversized databases that retains all activity rows for sampled entities; and (6) *source-overlap exclusion*, removing databases whose upstream source coincides exactly with that of a downstream evaluation benchmark.

**Phase 3 – Task proposal.**    We audit each database into a typed schema graph with two named roles: an *entity table* (whose primary key is referenced by an FK from another table) and an *activity table* (with a timestamp column and an FK to an entity table). A database is eligible for task generation if it has at least one of each, $\geq 5$ unique timestamps in the activity table, and $\geq 100$ rows. For each eligible database, an LLM (Claude Opus) proposes forecasting tasks via JSON, picking from 25 registered template families spanning eight reasoning regimes (Tab. 4): aggregation, threshold/existence, trend/change, behavior/churn, ranking, adoption/novelty, streak, and tail-event. A deterministic checker validates each proposal (column existence, FK roles, registered template); the LLM is run multiple times per database, with later passes biased toward under-represented templates and harder regimes (cold-start, rare-positive, heavy-tail).

**Phase 4 – Baselines and quality filtering.**    Each task is scored by two cheap historical baselines (entity-mean, last-value) and a fixed XGBoost over window-aggregation features (`n_estimators`=100, `max_depth`=6, `lr`=0.1, training capped at 5,000 rows; for the 20 most populous numeric columns, we compute count/mean/sum/max/min within a lookback of `5×timedelta`). Tasks fall into *learnable* (XGBoost beats the historical baselines), *random-tier* (signal within their noise— kept since stronger models may extract signal XGBoost cannot), *trivial*, *drift*, or *error*. We retain the first two and assign

coarse difficulty (easy/medium/hard) plus a fingerprint cluster that summarizes label-distribution and prediction-setting characteristics ($A$ rare-event cold-start, $B$ inverse imbalance, $C$ full cold-start, $D$ zero-inflated, $E$ heavy-tail, $F$ long-horizon churn, $G$ standard).

## D. Random-Walk Context Construction

For one task item, we score same-table candidate rows by random-walk visit counts (Stage 1), then consume the top-scoring candidates in order as launch points for the depth-bounded BFS cell collector BFSCOLLECT of the original RT, which we reuse unchanged (Stage 2). Walks step uniformly at random along PK and FK edges, restricted to neighbor rows whose timestamp does not exceed the target's timestamp $t^\star$; this enforces temporal validity at the walk level and avoids leaking future information into the context. Only visits to non-source rows in the target's table $\tau$ contribute to the score, so the resulting ordering ranks rows in the same table as the target by their personalized random-walk proximity.

---

**Algorithm 1** Random-walk context construction (one task item).

---

**Require:** target row $v^\star$ in table $\tau$ with timestamp $t^\star$; database $\mathcal{D}$; total cell budget $L$; per-seed cell budget $\ell$; BFS width $b$; recency flag $\rho \in \{0, 1\}$; walks $W$; walk length $K$.
**Ensure:** cell sequence $\mathbf{x}$ with $|\mathbf{x}| \leq L$.
 1: *Stage 1: score same-table candidates by random-walk visits.*
 2: $\text{score}[u] \leftarrow 0$ for every row $u$.
 3: **for** $w = 1$ **to** $W$ **do**
 4:    $u \leftarrow v^\star$.
 5:    **for** $k = 1$ **to** $K$ **do**
 6:       Move $u$ to a uniformly random PK/FK neighbor with timestamp $\leq t^\star$; if no such neighbor, end this walk.
 7:       **if** $u \neq v^\star$ **and** $u$ in table $\tau$ **then**
 8:          $\text{score}[u] \leftarrow \text{score}[u] + 1$.
 9:       **end if**
10:    **end for**
11: **end for**
12: *Stage 2: launch BFS from the top-scoring candidates.*
13: Order same-table rows by score descending; if $\rho=1$, break ties by timestamp descending.
14: $\mathbf{x} \leftarrow$ empty cell sequence.
15: **for** each row $s$ in this order **do**
16:    **if** $|\mathbf{x}| \geq L$ **then**
17:       **break**.
18:    **end if**
19:    Append BFSCOLLECT$(\mathcal{D}, s, \ell, b, t^\star, L - |\mathbf{x}|)$ to $\mathbf{x}$.
20: **end for**
21: **return** $\mathbf{x}$.

---

## E. RT vs. RDBLearn (Per-Task)

## F. Per-Ingredient and Per-Task Curves

Figs. 3 to 6 disaggregate the four ingredient ablations across the full context sweep $L$, and Figs. 9 and 10 give the per-task headline result. Per-task ablation curves follow the aggregate trends and are omitted for brevity.

## G. Per-Domain Corpus Breakdown

## H. Task Templates

## I. RelBench Forecasting Tasks

*Table 2.* Per-task RT (val-best $(\ell, w, \rho)$ at $L$=8k, 16-seed context-ensembled, $\sim$400 in-context labels) vs. RDBLearn (Zhang et al., 2026) trained on up to 10k labels per task (numbers from the KumoRFM-2 paper (Hudovernik et al., 2026)). Best per row in bold.

| DB | Task | RT ($\sim$400 labels) | RDBLearn (10k labels) |
|---|---|---|---|
| *Regression (nMAE %, lower is better)* | | | |
| rel-amazon | item-ltv | 10.26 | **8.22** |
| rel-amazon | user-ltv | 30.18 | **25.28** |
| rel-avito | ad-ctr | 48.42 | **35.53** |
| rel-event | user-attendance | 54.33 | **30.97** |
| rel-f1 | driver-position | **41.14** | 54.58 |
| rel-hm | item-sales | **11.73** | 12.92 |
| rel-stack | post-votes | **11.45** | 13.32 |
| rel-trial | site-success | **15.92** | 89.10 |
| rel-trial | study-adverse | 17.35 | **12.95** |
| **Mean** | | **26.75** | 31.43 |
| *Classification (AUROC %, higher is better)* | | | |
| rel-amazon | item-churn | 78.98 | **82.07** |
| rel-amazon | user-churn | **68.73** | 67.57 |
| rel-avito | user-clicks | 59.14 | **69.04** |
| rel-avito | user-visits | 61.96 | **65.49** |
| rel-event | user-ignore | **85.58** | 82.52 |
| rel-event | user-repeat | **76.80** | 75.04 |
| rel-f1 | driver-dnf | **82.49** | 70.87 |
| rel-f1 | driver-top3 | **91.70** | 79.69 |
| rel-hm | user-churn | 67.13 | **68.05** |
| rel-stack | user-badge | 75.25 | **85.26** |
| rel-stack | user-engagement | 83.94 | **89.39** |
| rel-trial | study-outcome | 66.46 | **71.58** |
| **Mean** | | 74.85 | **75.97** |

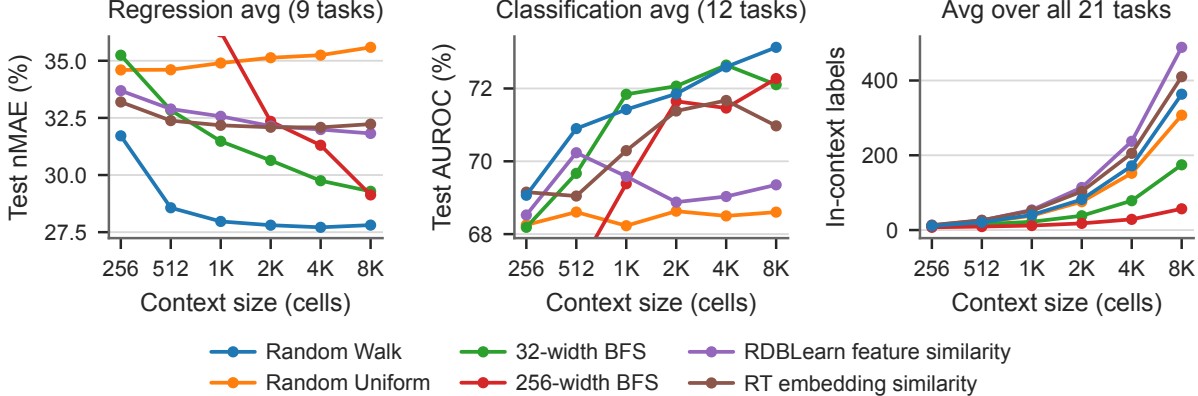

*Figure 3.* Retriever ablation. Random walk (blue) is best on regression at every $L$ and on classification at $L$=8k. Higher label yield (right panel) does not imply better accuracy: RDBLearn-feature similarity delivers more labels yet trails by $\sim$4 nMAE.

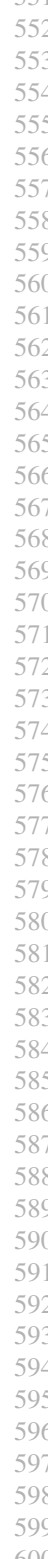

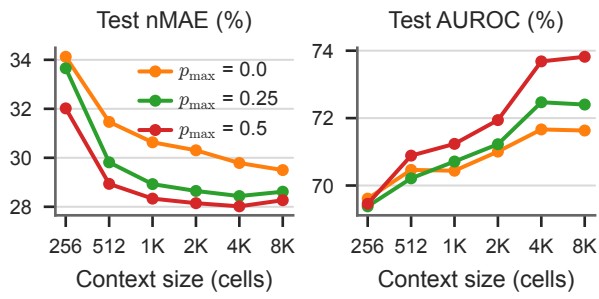

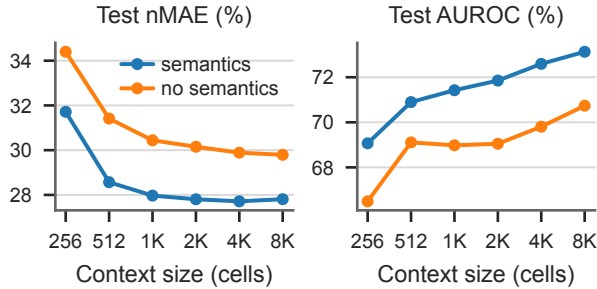

*Figure 4.* Multi-cell masking. $p_{max}=0$ recovers the original RT (Ranjan et al., 2026). At $L=8k$, $p_{max}=0.5$ leads $p_{max}=0$ by 4.2% nMAE and 3.1% AUROC.

*Figure 5.* Schema-semantics ablation. Shuffling table and column names degrades both metrics by 6.7%–9.1% nMAE and 2.6%–4.1% AUROC across the full context sweep.

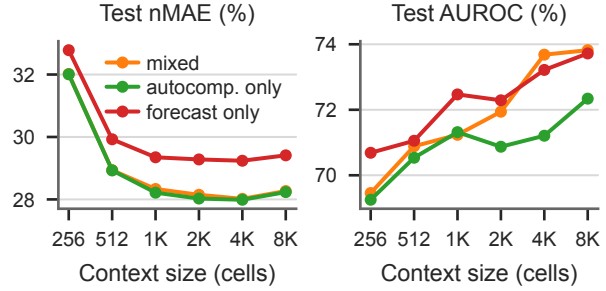

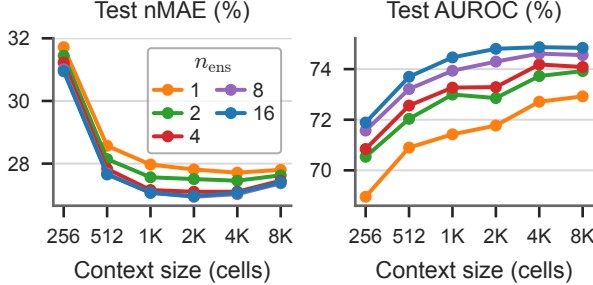

*Figure 6.* Pretraining task-mix. Each single-task variant specializes to one metric; only mixed is competitive on both.

*Figure 7.* Context ensembling at test time. Up to 3.2% nMAE and 4.3% AUROC gains across $L$.

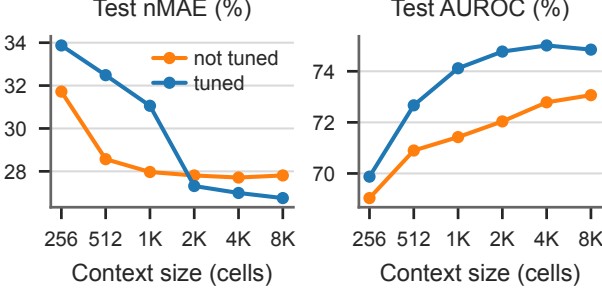

*Figure 8.* Per-task context tuning. Tuning improves classification at every $L$, and regression once $L$ is large enough to host the chosen $\ell$.

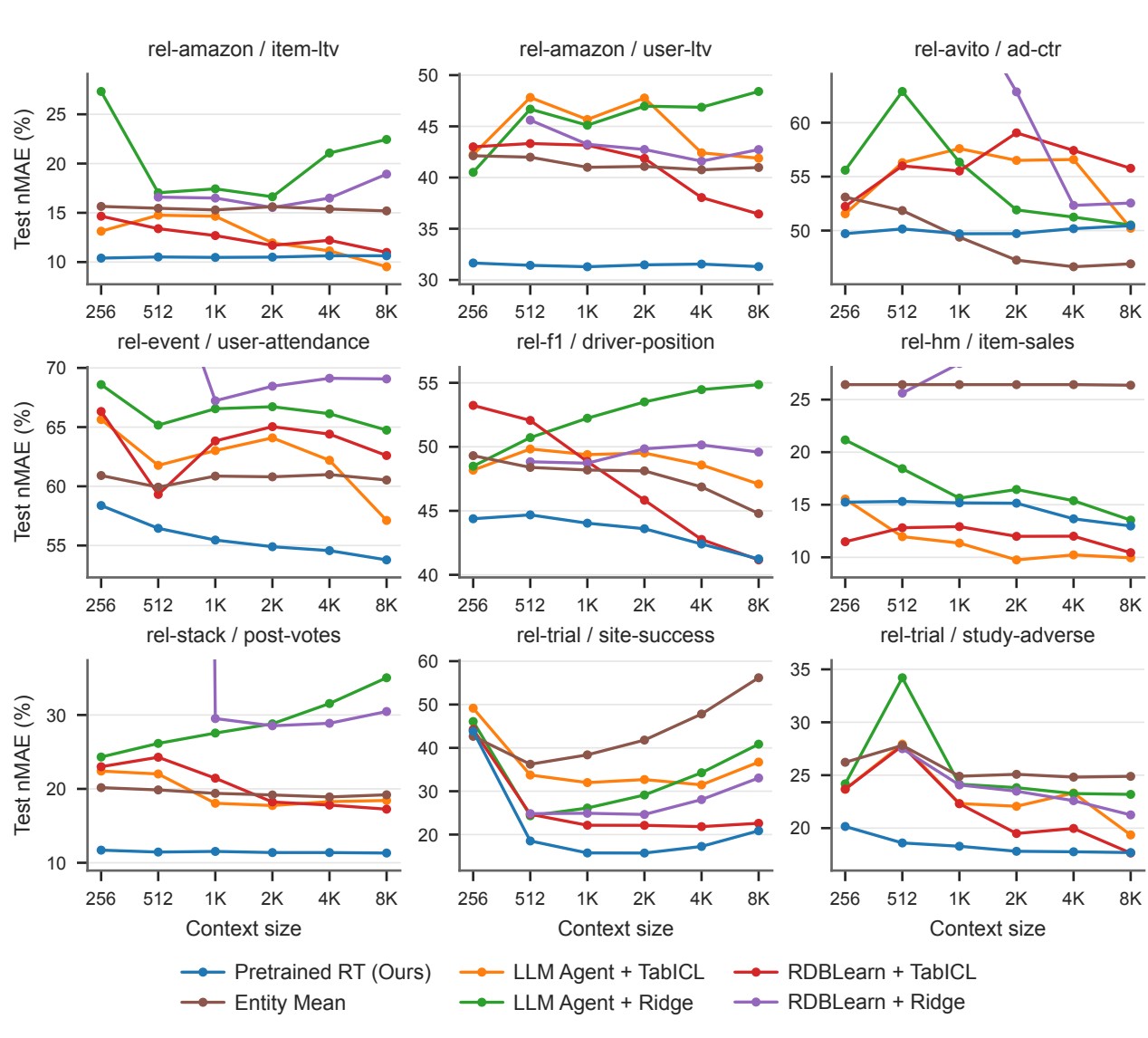

*Figure 9.* Per-task test MAE versus inference-time context size on the nine RelBench regression tasks. Lower is better.

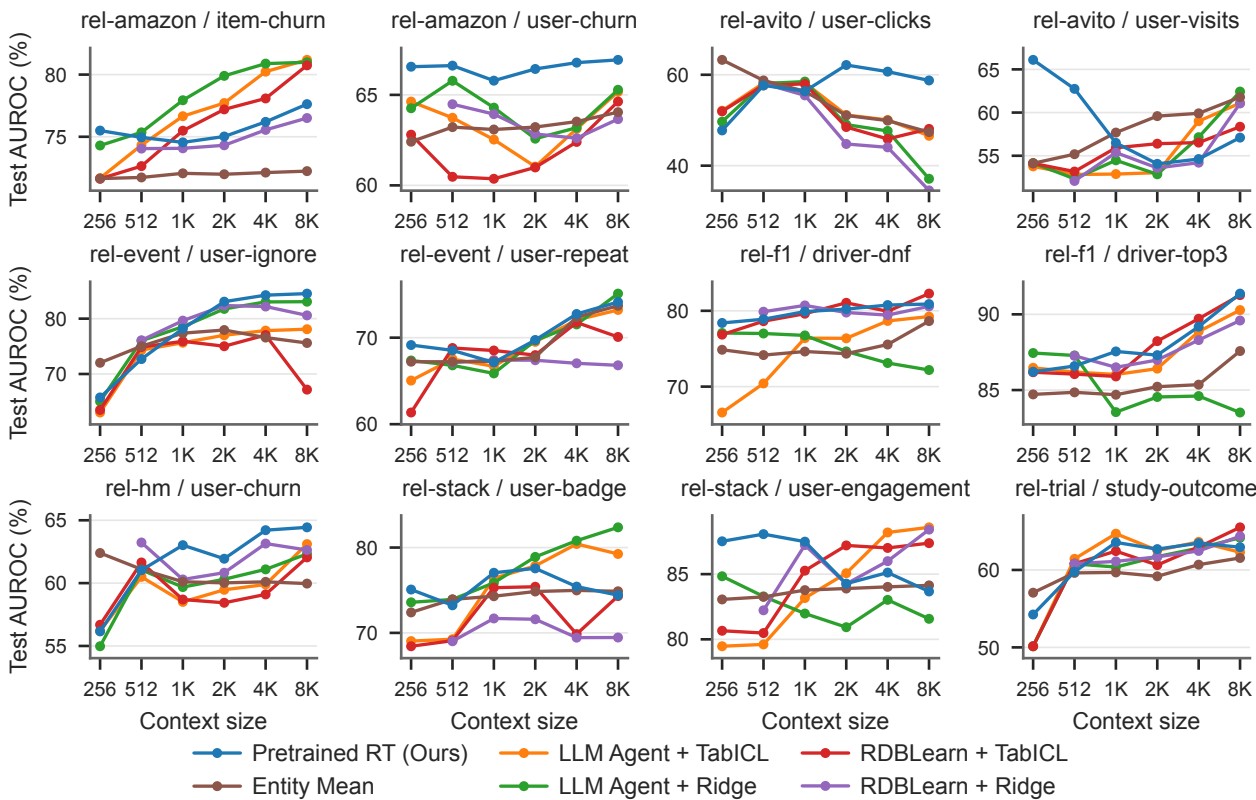

*Figure 10.* Per-task test AUROC versus inference-time context size on the twelve RelBench classification tasks. Higher is better.

*Table 3.* Per-domain corpus breakdown across all 63 collected domains, grouped by italic subject-area headers and sorted by descending DB count. *DBs*: raw count; *Tasks*: count after eligibility, audit, and quality filtering (dash = none yielded); *Tabl./Cols/Rows/Span*: per-domain medians of structural shape.

| Domain | Source | DBs | Tasks | Tabl. | Cols | Rows | Span |
|---|---|---|---|---|---|---|---|
| *Commerce & Online Platforms* | | | | | | | |
| tech community | Stack Exchange dumps | 76 | 2,277 | 7 | 51 | 874K | 13.2y |
| recommendations | Public e-commerce dumps | 36 | 1,217 | 3 | 16 | 617K | 16.5y |
| prediction market | PredictIt / Polymarket | 2 | 63 | 4 | 26 | 617K | 1.8y |
| ecommerce | Public e-commerce dumps | 2 | 59 | 4 | 13 | 5.6M | 159d |
| consumer product | Curated CPG datasets | 1 | 52 | 4 | 17 | 349K | 6.0y |
| food reviews | Yelp / OpenTable | 1 | 11 | 3 | 13 | 679K | 19.4y |
| real estate | Public listing dumps | 1 | — | 2 | 4 | 383K | — |
| *Sports, Culture & Media* | | | | | | | |
| motorsport | Ergast / FIA archives | 9 | — | 2 | 19 | 4.8K | 24.0y |
| sports | Lahman / ATP / StatsBomb / Cricsheet | 6 | 164 | 5 | 49 | 315K | 23.9y |
| food | USDA / Open Food Facts / RateBeer | 5 | 158 | 3 | 15 | 1.7M | 13.6y |
| music | MusicBrainz / Million Song / Last.fm | 3 | 35 | 4 | 14 | 143K | 31.7y |
| museum | Museum collection APIs | 3 | 25 | 3 | 40 | 353K | 143.2y |
| gaming | Game telemetry dumps | 3 | 21 | 3 | 19 | 3.8K | 3.4y |
| cultural | Cultural-heritage portals | 2 | — | 3 | 16 | 83K | — |
| cultural surveys | ICPSR / GESIS | 2 | — | 5 | 33 | 684K | 16.5y |
| podcast | Podcast directory APIs | 1 | 29 | 3 | 7 | 554K | 210d |
| history | Historical archives | 1 | — | 3 | 15 | 2.2K | 1.9y |
| *Government, Finance & Energy* | | | | | | | |
| government us | US Treasury / SEC EDGAR / CFPB | 11 | 121 | 2 | 54 | 748K | 3.6y |
| energy | EIA / OPSD | 6 | 73 | 3 | 158 | 217K | 59.7y |
| international | World Bank / IMF | 4 | 47 | 3 | 18 | 46K | 32.5y |
| finance | Stooq / public market dumps | 2 | 59 | 2 | 14 | 159K | 17.8y |
| politics | FEC / OpenSecrets | 2 | 32 | 2 | 6 | 2.5M | — |
| eia forms | EIA Form-861 / 923 | 2 | 6 | 3 | 141 | 41K | 70.5y |
| civic | Civic data portals | 1 | 29 | 4 | 19 | 501K | 46d |
| energy opsd | Open Power System Data | 1 | 17 | 2 | 5 | 602K | 2.8y |
| crypto | CoinGecko / blockchain dumps | 1 | — | 3 | 31 | 20K | 56.3y |
| economics | FRED / OECD | 1 | — | 3 | 16 | 99K | 23.0y |
| energy infrastructure | FERC / pipeline registries | 1 | — | 4 | 22 | 1.1K | 69.5y |
| labor | BLS / OECD | 1 | — | 2 | 12 | 300K | — |
| legal | CourtListener / SCOTUS | 1 | — | 4 | 20 | 933K | 6.7y |
| patent | USPTO / PatentsView | 1 | — | 4 | 25 | 125K | 18.0y |
| uk | UK gov data portals | 1 | — | 4 | 31 | 40K | — |
| *Health & Public Services* | | | | | | | |
| clinical | ClinicalTrials.gov / AACT | 5 | 27 | 3 | 29 | 150K | 22.0y |
| biology | NCBI / UniProt | 4 | — | 2 | 21 | 122K | 31.1y |
| healthcare quality | CMS Hospital Quality | 2 | 37 | 4 | 125 | 248K | 4.7y |
| education | NCES | 1 | 43 | 7 | 56 | 11M | 3.9y |
| public health | CDC / WHO | 1 | 12 | 3 | 34 | 53K | — |
| transport | GTFS / NYC TLC | 1 | 8 | 7 | 52 | 585K | — |
| agriculture | USDA / FAO | 1 | — | 3 | 41 | 1.4M | — |
| biology advanced | ENCODE / GTEx | 1 | — | 3 | 21 | 9.7M | — |
| disaster | FEMA / EM-DAT | 1 | — | 2 | 12 | 6.2K | 72.9y |
| skills | O*NET / job-posting dumps | 1 | — | 10 | 94 | 354K | 19.7y |
| *Scientific & Reference* | | | | | | | |
| text2sql | Spider / BIRD benchmarks | 260 | 487 | 5 | 25 | 79.5 | 12.0y |
| ctu | CTU Prague Repository | 118 | 407 | 4 | 33 | 29K | 17.6y |
| temporal network | SNAP / KONECT | 12 | 457 | 2 | 6 | 579K | 3.3y |
| network | SNAP / KONECT (raw) | 10 | — | 2 | 3 | 1.0M | 12.6y |
| academic | OpenAlex / DBLP | 7 | 92 | 4 | 27 | 224K | 6.9y |
| geospatial | OpenStreetMap / GADM | 6 | 98 | 3 | 13 | 3.5M | 1.7y |
| graph learning | OGB / DGL benchmarks | 5 | — | 3 | 8 | 1.2M | 49.0y |
| fivethirtyeight | FiveThirtyEight datasets | 4 | — | 2 | 45 | 1.5K | 39.3y |
| ecology | GBIF / iNaturalist | 3 | 34 | 3 | 47 | 110K | — |
| linguistics | LDC / UD treebanks | 3 | — | 4 | 30 | 32K | — |
| security | MITRE / CVE feeds | 3 | — | 5 | 36 | 79K | 8.5y |
| astronomy | Sloan / Gaia | 2 | — | 2 | 269 | 355K | — |
| geonames | GeoNames | 1 | 30 | 3 | 43 | 34K | 20.2y |
| forecasting | M-competitions / Rossmann | 1 | 23 | 3 | 11 | 302K | 1.3y |
| hydrology | USGS / NOAA | 1 | 5 | 3 | 15 | 4.4K | 364d |
| biodiversity | GBIF / IUCN | 1 | — | 3 | 32 | 21K | 305d |
| environment | EPA / Copernicus | 1 | — | 3 | 69 | 85K | — |
| network infra | RIPE / CAIDA | 1 | — | 5 | 22 | 2.0K | 2.2y |
| ocean | NOAA / Argo | 1 | — | 2 | 22 | 2.1K | 45d |
| storm | NOAA Storm Events | 1 | — | 3 | 44 | 125K | 364d |
| threat intel | AlienVault / VirusTotal | 1 | — | 2 | 10 | 281K | 7.8y |
| **Total** | | **650** | **6,255** | | | | |

*Table 4.* The 25 task-template families instantiated by the corpus, organized under italic signal-type headers. Each row gives the prediction target as the model sees it, paired with one concrete domain example. R: regression; C: binary classification.

| Template | Type | Prediction target (example) |
|---|---|---|
| *Aggregation — volume / intensity of activity* | | |
| aggregate_numeric | R | mean / sum of a numeric column over the next window *(e.g. a player's home runs next season)* |
| count_events | R | number of events for the entity in the next window *(e.g. orders per customer next month)* |
| distinct_count | R | distinct values of a column the entity touches *(e.g. distinct co-authors next year)* |
| concentration | R | entity's share of total activity *(e.g. seller's share of category sales)* |
| *Threshold / existence — continuous signal at a decision boundary* | | |
| threshold_binary | C | whether an aggregate exceeds a threshold *(e.g. revenue $>$\$1M next quarter)* |
| existence_binary | C | whether the entity appears in the event table next window *(e.g. user posts at least once)* |
| top_k_binary | C | whether the entity is in the top $K\%$ on a metric *(e.g. artist enters top 1% of plays)* |
| *Trend / change — direction and magnitude of change* | | |
| trend_binary | C | whether a metric rises relative to the current window *(e.g. weekly volume rises)* |
| trend_reversal | C | whether the direction of change flips between windows *(e.g. sales reverse from rising to falling)* |
| change_numeric | R | numeric delta from current to next window *(e.g. change in monthly energy consumption)* |
| volatility_numeric | R | standard deviation of a metric in the next window *(e.g. next-week price volatility)* |
| *Behaviour / churn — entity lifecycle* | | |
| churn_activity | C | entity has *no* activity in the next window *(e.g. subscriber cancels next month)* |
| return_binary | C | dormant entity reactivates next window *(e.g. dormant Stack Exchange user posts again)* |
| time_until_next | R | days until the entity's next event *(e.g. days until a player's next ATP match)* |
| will_repeat_last | C | next event's category equals the most-recent one *(e.g. customer re-buys the same product)* |
| first_event_binary | C | entity experiences a first occurrence next window *(e.g. first purchase by a new account)* |
| *Ranking — relative position within a peer group* | | |
| rank_change | C | entity's rank within its group improves by $\geq K$ *(e.g. driver climbs $\geq$5 places)* |
| rank_regression | R | numeric rank within the group (1 = best) *(e.g. predict the driver's next-race finish position)* |
| peer_outperform | C | entity beats the mean of its peer group *(e.g. artist beats peer-group median plays)* |
| *Adoption / novelty — first-time events* | | |
| new_category_adoption | C | entity touches a category value it never has before *(e.g. first purchase from a new product category)* |
| derived_ratio | R | ratio of two numeric columns over the next window *(e.g. new-to-returning customer ratio)* |
| *Streak — resilience as a single metric* | | |
| streak_length | R | longest run of consecutive active sub-periods *(e.g. consecutive seasons a player remains active)* |
| *Tail-event (hard-tier) — rare and extreme regimes* | | |
| cold_start_binary | C | newly-appearing entity has repeat activity in its first window *(e.g. new user posts within 24h of joining)* |
| rare_positive_binary | C | rare event ($\leq 10\%$ positive rate) occurs for the entity *(e.g. rare adverse-quality hospital event)* |
| heavy_tail_regression | R | $\log_1 p$ of an aggregate; preserves heavy upper tail *(e.g. $\log_1 p$ of next-month spend, large spenders preserved)* |

*Table 5.* Binary classification forecasting tasks in RelBench.

| Dataset | Task | Entity | Prediction target |
|---|---|---|---|
| rel-amazon | user-churn | User | Whether the user does not review any product in the next 3 months. |
| rel-amazon | item-churn | Item | Whether the item receives no reviews in the next 3 months. |
| rel-avito | user-visits | User | Whether the user visits more than one ad in the next 4 days. |
| rel-avito | user-clicks | User | Whether the user clicks more than one ad in the next 4 days. |
| rel-f1 | driver-dnf | Driver | Whether the driver does not finish a race in the next month. |
| rel-f1 | driver-top3 | Driver | Whether the driver qualifies in the top 3 for a race in the next month. |
| rel-hm | user-churn | User | Whether the customer makes no transactions in the next week. |
| rel-stack | user-engagement | User | Whether the user makes any votes, posts, or comments in the next 3 months. |
| rel-stack | user-badge | User | Whether the user receives a new badge in the next 3 months. |
| rel-trial | study-outcome | Study | Whether the trial achieves its primary outcome in the next year. |
| rel-event | user-repeat | User | Whether a previously active user attends another event in the next 7 days. |
| rel-event | user-ignore | User | Whether the user ignores more than two event invitations in the next 7 days. |

*Table 6.* Regression forecasting tasks in RelBench.

| Dataset | Task | Entity | Prediction target |
|---|---|---|---|
| rel-amazon | user-ltv | User | Value of the products the user buys and reviews in the next 3 months. |
| rel-amazon | item-ltv | Item | Value of the purchases and reviews the item receives in the next 3 months. |
| rel-avito | ad-ctr | Ad | Click-through rate for the ad over the next 4 days. |
| rel-f1 | driver-position | Driver | Average finishing position of the driver across races in the next 2 months. |
| rel-hm | item-sales | Item | Total sales for the article in the next week. |
| rel-stack | post-votes | Post | Number of votes the post receives in the next 3 months. |
| rel-trial | study-adverse | Study | Number of affected patients with severe adverse events in the next year. |
| rel-trial | site-success | Site | Success rate of a trial site in the next year. |
| rel-event | user-attendance | User | Number of events the user responds yes or maybe to in the next 7 days. |

