# OpenReview forum: "Large-Scale Pretraining unlocks Few-Shot Prediction for Relational Data"
_ICML.cc/2026/Workshop/FMSD — FMSD @ ICML 2026 Poster_

### Official Review · Reviewer_MztM · 2026-05-20

**Rating:** 8
**Confidence:** 4

**Review:**

# Summary

This paper proposes a few-shot pretraining recipe for Relational Transformers that enables strong in-context learning on multi-table relational databases. The key idea is that the few-shot capability of existing tabular foundation models can be significantly improved by redesigning the pretraining data and objectives for relational settings. To this end, the paper introduces a new large-scale relational pretraining dataset, THE JOIN, together with training strategies such as multi-cell masking, mixed context-size training, and random-walk-based retrieval. Experiments on RelBench demonstrate strong label efficiency and competitive or superior performance compared to existing in-context relational/tabular baselines and even fully supervised methods. The paper also includes extensive ablation studies analyzing the contribution of each component.

# Strengths

- The paper addresses an important and timely problem: enabling strong few-shot learning for relational data without task-specific training.
- The proposed training recipe is well-motivated and technically coherent, especially the alignment between pretraining and inference-time few-shot behavior.
- Introducing THE JOIN is a meaningful contribution that can benefit future research on relational foundation models.
- The random-walk-based retrieval and multi-cell masking strategies are intuitive and empirically effective.
- The experiments are comprehensive, including careful ablation studies and label-efficiency analyses.
- The paper is generally clear and well-organized, and the empirical gains on RelBench are convincing.

# Weaknesses

- The paper lacks comparisons with several recent relational foundation models (e.g., KumoRFM, Griffin), which would better position the contribution within the broader literature.
- Runtime and memory efficiency are insufficiently discussed, especially given the large context sizes and ensemble-based inference settings.
- Some ablations are conducted with reduced training budgets, making it unclear whether the same conclusions hold at full convergence.
- While the proposed retrieval strategy is effective, its computational overhead is not carefully quantified.

# Overall Assessment

Overall, I found this paper to be strong and impactful. The work demonstrates that carefully designed pretraining objectives and datasets can substantially improve few-shot in-context learning for relational transformers, surpassing existing tabular ICL approaches. The introduction of THE JOIN and the thorough empirical evaluation further strengthen the contribution. Although additional comparisons and efficiency analyses would improve the paper, the current results are already compelling and likely to be influential for future relational foundation model research. I recommend acceptance.

---

### Official Review · Reviewer_SMdk · 2026-05-21
**A comprehensive framework for few-shot prediction on relational data**

**Rating:** 8
**Confidence:** 3

**Review:**

Summary

This paper introduces a framework to pretrain a foundation model for few-shot predictions on relational databases. The main contributions include the introduction of a new pretraining dataset, a training recipe that enables few-shot prediction, and context ensembling and tuning strategies during test time.

Strengths

1.	Dataset Contribution: Provides a novel addition to existing pretraining data for relational databases.
2.	Pretraining Innovations: The combination of mixed context sizes, random-walk retrieval, and multi-cell masking offers pragmatic solutions for pretraining a model capable of few-shot and long-context inference.
3.	Strong Performance: The proposed approach outperforms all baselines, including a model trained on the full task.

Areas for Improvement

1.	Use of context ensembling and tuning: The baselines do not appear to benefit from test-time context tuning. Because this tuning is performed on a validation set—which the other approaches might not utilize—there may be a dataset mismatch. To ensure fairness, the baselines should potentially be provided with both the training and validation labels.

Detailed Comments

1.	Is it possible to implement a test-time context ensembling or tuning strategy for the other baselines?
2.	What are the test-time compute requirements and time differences compared to the fully supervised baseline, RelGNN?

Justification of Score

This paper makes strong contributions to the task of few-shot relational database prediction. It provides a comprehensive framework for pretraining, including a new dataset and novel recipes for both pretraining and inference time.

---

### Official Review · Reviewer_mTCE · 2026-05-21
**Review of the paper - Large-Scale Pretraining unlocks Few-Shot Prediction for Relational Data**

**Rating:** 6
**Confidence:** 3

**Review:**

**Summary**- This paper addresses the following gap - while foundation models for language and vision have achieved strong few-shot adaptation, relational database prediction still typically requires thousands of labeled examples and task-specific training. The paper argue that the bottleneck is not architectural but lies in the pretraining recipe, and they propose three fixes on top of the Relational Transformer (RT) backbone - (a) THE JOIN, a new large-scale pretraining corpus of 6,255 forecasting tasks across 650 real-world databases spanning 63 domains; (b) a revised pretraining procedure combining mixed context sizes, multi-cell masking, and a random-walk-based context retriever that prioritizes topologically close and label-bearing rows; and (c) test-time compute scaling via context ensembling and per-task context tuning. Evaluated on the RelBench benchmark, the resulting model matches the strongest in-context learning pipelines (RDBLearn + TabICLv2 and LLM Agent + TabICLv2) using 23-32 times fewer in-context labels and exceeds the prior fully-supervised state of the art (RelGNN) without any task-specific training. Ablations in Section 7 isolate the contribution of each recipe ingredient.

**Strengths**
1. The paper clearly motivates why prior relational foundation models fail in the few-shot regime - narrow pretraining data, short fixed context windows, and naive breadth-first-search context construction. This maps directly onto the three contributions, giving the paper a nice internal coherence, making it easy to read and follow.

2. THE JOIN is a significant open-data contribution - 650 real-world databases from 63 domains, 6,255 tasks generated via an LLM-assisted pipeline with deterministic validation and XGBoost-based quality filtering. The pipeline is carefully engineered - source-overlap exclusion to prevent leakage into RelBench, deterministic checker validation, and deliberate coverage of rare-event/cold-start/heavy-tail regimes.

3. The paper presents a useful ablation study over the main training and retrieval ingredients while also separately analyzing test-time context tuning and ensembling.

4. The work presents rigorous test-time compute analysis. These are very useful practical results for deployment. Moreover, the empirical results are strong with a decent number of baselines.

**Weaknesses**

1. Classification gains are noticeably more modest than regression gains. It reports up to 18% relative nMAE improvement on regression but only 3% relative AUROC improvement on classification over the next-best baseline. Table 1 also shows that the full recipe reaches 73.1 AUROC at L=8k, increasing to 74.8–74.9 only with context ensembling or per-task tuning. The paper notes that classification continues improving at L=8192, while regression plateaus around L≈512, but does not deeply analyze why these scaling behaviors differ.

2. No wall-clock inference times are provided anywhere in the paper. This matters practically, an LLM Agent with Claude Opus 4.6 writing SQL queries may be far slower than RT at L=256, but RT at L=8k with n_ens=16 may be slower than the LLM Agent. Without latency numbers, the efficiency claim in the abstract is incomplete.

3. Dependence on frozen LM embeddings for schema semantics is not adequately analyzed. it does show that schema semantics contribute 6.7% - 9.1% nMAE and 2.6% - 4.1% AUROC. However, the ablation only compares full schema semantics against randomly shuffled names, it does not study which LM is used for the frozen text embeddings, whether fine-tuning those embeddings would help, or how the model behaves on databases with uninformative column names.

4. Since KumoRFM/KumoRFM-2 are described as closely related schema-agnostic relational foundation models for in-context learning, the absence of a direct KumoRFM-2 comparison makes the claim, "beats every in-context baseline", difficult to fully assess. At minimum, the paper should explain whether KumoRFM-2 is not available or not comparable under the same protocol.